# Non-Contact Vision-Based Techniques of Vital Sign Monitoring: Systematic Review

**DOI:** 10.3390/s24123963

**Published:** 2024-06-19

**Authors:** Linas Saikevičius, Vidas Raudonis, Gintaras Dervinis, Virginijus Baranauskas

**Affiliations:** Automation Department, Faculty of Electrical and Electronics Engineering, Kaunas University of Technology, 44249 Kaunas, Lithuania; linas.saikevicius@ktu.edu (L.S.); gintaras.dervinis@ktu.lt (G.D.); virginijus.baranauskas@ktu.lt (V.B.)

**Keywords:** non-contact measurements, remote sensors, image processing, vital signs, photoplethysmography

## Abstract

The development of non-contact techniques for monitoring human vital signs has significant potential to improve patient care in diverse settings. By facilitating easier and more convenient monitoring, these techniques can prevent serious health issues and improve patient outcomes, especially for those unable or unwilling to travel to traditional healthcare environments. This systematic review examines recent advancements in non-contact vital sign monitoring techniques, evaluating publicly available datasets and signal preprocessing methods. Additionally, we identified potential future research directions in this rapidly evolving field.

## 1. Introduction

In the past 15 years, advancements in camera technology have coincided with increased availability and affordability, leading to an increasing interest in using these technologies in healthcare settings. Image-based monitoring methods can simultaneously measure multiple vital signs using a non-contact sensor. Imaging photoplethysmography (iPPG) is an optical technique that uses a simple camera to assess several vital functions, such as heart rate and respiratory rate. Researchers have made significant efforts to reliably estimate heart and respiratory rates. Currently, research is focusing on the remote estimation of pulse, respiratory rate, oxygen saturation and blood pressure (BP). While there is an increasing number of articles and research on pulse and respiratory rate monitoring, there is also a limited number of publicly available publications on advancements related to BP estimation. The most monitored vital signs using non-invasive methods are heart rate, temperature, respiratory rate and oxygen saturation. Not so common and under development is blood pressure.

There is an estimated dependency of the pulse amplitude of the heart cycle. A suggestion is that depending on the blood pressure, there is a different length in systolic upstroke and diastolic time and also pulse amplitude width. In theory, it should be possible to estimate blood pressure from these key timings and amplitudes by adding blood pressure to remote the vital sign monitoring domain. While pulse and respiratory rate are primary health measurement data providers, blood pressure is a deep and more informative health status. High blood pressure needs to be detected in the early stages as it shows no symptoms until heart disease or failure occurs or even lethal outcomes. Annually, high blood pressure causes approximately 12.8% of deaths worldwide and 3.7% of disabilities that are considered permanent [1]. A multitude of studies have provided confirmation that this specific factor constitutes a substantial risk for the onset of numerous cardiovascular ailments, including but not limited to coronary heart disease (CHD), ischemic heart disease, atherosclerosis, myocardial infarction (MI) and hemorrhagic stroke.

Throughout the 20th century, researchers made significant strides in improving the sphygmomanometer, such as enhancing diastolic pressure readings and increasing device durability [2]. Newer manometers successfully addressed the accuracy issues that plagued their mid-century predecessors, producing precise measurements of mean arterial pressure (MAP). However, over time, these automated oscillometric manometers also encountered issues, causing significant discomfort for children, the elderly and those with medical challenges [2]. Using infrared (IR) light, photoplethysmography (PPG) can measure changes in blood vessel volume and provide valuable information about cardiovascular health. It can even accurately estimate blood pressure changes. Typically, PPG devices have a light source and sensor and measure reflected light in response to changes in blood volume [3]. Depending on the region of application, IR or light-emitting diode (LED) light sources are usually used.

Multiple past reviews have concluded that various methods of measuring blood pressure (BP) often require invasive devices or cumbersome equipment, which can be inconvenient to carry around. A lack of annotated training data was identified as a key application issue of machine learning-based methods. To address this issue, researchers have explored the idea of measuring BP in a non-contact way. One promising solution is remote photoplethysmography (rPPG), which utilizes color digital camera to capture subtle changes in light reflected from human skin. This paper seeks to review the latest deep learning applications in the non-contact monitoring of vital signs that are applicable in a the context of a home environment and can be used with a conventional device, such as, color cameras. This article is organized as follows. Section 2 presents the materials and methodology used for the scientific article search and selection. Selected articles are briefly reviewed in Section 3. Contextual analysis is performed in Section 4 and the article is concluded with a discussion of the findings in Section 5.

## 2. Materials and Methods

This study was conducted using Preferred Reporting Items for Systematic Reviews and Meta-Analyses (PRISMA) guidelines. The following section provides a detailed explanation of this article’s search procedure.

### 2.1. Eligibility Criteria

Articles were selected based on the qualifying requirements listed in Table 1. Key factors included quality, accessibility, comparability and methodological clarity. Metrics such as the publishing language, paper type, complete paper availability, medical domain, goal, use of data modalities, deep learning use and performance assessment were included in the criteria. Articles were filtered using exclusion criteria based on the title, abstract and keywords, followed by being thoroughly reviewed to assess their suitability.

### 2.2. Article Search Process

Primary and secondary search tactics were employed using electronic databases such as Scopus and Web of Science. The primary search was conducted on the 25th of March 2024 and considered articles published had to have been published since 2019. The most significant abbreviations (such as RPPG, PPG, HR) were incorporated in Boolean logic queries that were defined by concatenating words using the OR gate and combining term categories using the AND gate. Several phrases (such as Econ, EART and so on) were used in order to include or exclude research papers and area symbols and guide the search results. The search query was limited to the abstract and title.

The whole search code used was the following:

((“blood pressure”) AND (“remote”) AND (“vital”) AND (“heart rate”) AND (“video”) AND (“photoplethysmography”) AND (rppg)) AND PUBYEAR > 2016 AND PUBYEAR < 2024 AND (EXCLUDE (SUBJAREA, “VETE”) OR EXCLUDE (SUBJAREA, “ARTS”) OR EXCLUDE (SUBJAREA, “AGRI”) OR EXCLUDE (SUBJAREA, “BUSI”) OR EXCLUDE (SUBJAREA, “CENG”) OR EXCLUDE (SUBJAREA, “CHEM”) OR EXCLUDE (SUBJAREA, “IMMU”) OR EXCLUDE (SUBJAREA, “ECON”) OR EXCLUDE (SUBJAREA, “PHAR”) OR EXCLUDE (SUBJAREA, “EART”) OR EXCLUDE (SUBJAREA, “PSYC”) OR EXCLUDE (SUBJAREA, “DECI”)) AND (LIMIT-TO (LANGUAGE, “English”)) AND (LIMIT-TO (DOCTYPE, “ar”) OR LIMIT-TO (DOCTYPE, “cp”) OR LIMIT-TO (DOCTYPE, “ch”)).

Excluded areas included the fields of Veterinary Medicine, Arts, Agriculture, Business, Economics, Pharmacology and Psychology. We looked at the final stage of each selected paper. Some papers were wrongly named, where instead of the article there was some commercial like for FLIR cameras or a World Health Organization overview. References discovered in articles from the initial search were examined again and manually included during the secondary search, contingent upon their relevance as determined by the eligibility criteria.

### 2.3. Selection Process

Selection was made from the Scopus and IEEE databases, the main articles were taken from Scopus, although many were found in the IEEE. The title, authors, publication date, paper type, article venue, complete abstract and keywords were among the search data. Duplicate articles were removed using the Microsoft^®^ 365 Excel tool and checking for repetitive titles. Excluded articles also included those that were not in journals or in the English language. The remaining publications’ titles, abstracts and keywords were filtered using the standards listed in Table 1. After reading every article, those that did not fit the requirements were eliminated. A PRISMA flow diagram is presented in Figure 1. 

### 2.4. Selection Summary

The type of data was determined by the review’s objectives. Some article metadata could be found in online databases, but the following information was extracted during a full article read in order to quantitatively evaluate the articles: objectives, subject and classes, methodology, initial sample count, training sample count, preprocessing and machine learning methods. Regretfully, several publications failed to provide the necessary information. Consequently, assumptions regarding the absent data were made as the following:Number of training samples—If this number was not provided, the initial sample count was taken into account when evaluating the dataset location;Application of preprocessing techniques—It is deemed that none were applied.

## 3. Results

When searching the Scopus and Web of Science databases 1024 articles were found. After duplicate articles were removed, 948 original articles remained. Titles, abstracts and keywords were compared to the qualifying criteria in this group (Figure 1). The entire texts of the remaining 948 articles were assessed using the same set of criteria. In the end, 110 articles were chosen, and 26 more were added as a consequence of forward snowballing and reference reviews in related review articles. This section displays the analytical results for the articles listed below. The results of the searches are represented in Figure 2, which shows the distribution of articles by date. It is clear that interest in this area of study has been growing for the previous five years.

It is obvious and mentioned in most of the articles that this field of interest became needed more and more with the COVID-19 pandemic to monitor vital signs without primary contact with the patient. Today, as modern society gets older, it is needed to monitor elderly people from their homes. Contact measurements mostly have some discomfort or body intrusion elements, while contactless ones can even be unnoticed. Most publications on the subject are coming from China and the United States, as shown in Figure 3.

Studies are made mostly in Computer Science and Engineering, as remote vital sign monitoring includes visual data interpretation in mathematical ways to obtain and check differences between arrays of pixels and colors in taken pictures or video streams. All this data must be mathematically, or with the help of AI, summarized and presented to the user in a simple number or trend format. Also, data must be accurate because they are related to human health. To make data more accurate data from medicine were used. One of the datasets entailed premade video streams taken during different human stress and calm states with information on the heartbeat taken by a professional, medical-grade HR monitoring device. The Physics and Astronomy subject area is shown as separate, but uses the same methods as Engineering and Computer science—data gathering, signal processing as RGB or motion analysis processed with different mathematical algorithms. The detailed articles spread by their area are represented in Figure 4.

From representation of results in Figure 2, Figure 3 and Figure 4 there were excluded other systematic reviews like [123,124,125,126,127].

Connected Papers exploration is a technique used to discover and navigate the academic literature in the field of non-contact vital sign monitoring. Paper exploration started from key research articles in the research field. The citation tracking tool was used to find other related research papers to the initial one. Finally, a connection network of research was created that explored the development and advancement in non-contact vital sign monitoring technologies. This exploration is helping to understand the current state of the field, identify emerging trends and see how different research groups are contributing to the technology. Connected Papers exploration is based on a starting article written by Wenjin Wang, “Algorithmic Principles of Remote PPG”, whose principles were cited 677 times in the following years after publication. It is obvious that subequal articles one way or another use a primary one as the start state and add their own approaches and variations. How articles reference each other, or their relations, are shown in Figure 5.

To categorize and organize the information from articles, a Systematic Literature Review (SLR) coding scheme was used. It is applied to the detailed sorting and labeling of a large amount of information from academic papers. Firstly, categories and groups of information like the vital signs, technology and devices used, patient experience during examination and feasibility of the method were decided upon. Then, articles were sorted according to these groups and categories. Once articles were sorted, information in it like a paragraph, meaning and method was labeled as belonging to the appropriate category. Doing this helped to sort and analyze the data further on. Statistical analysis was also conducted. For this categorization and labeling, the MaxQDA tool was used. Important in creating a coding scheme was separating non-contact methods and looking for ways and methodologies to have non-contact or remote vital sign monitoring, including by considering the following factors: what studies were made; how simple the method was; what their feasibility was; how practical the method was when applied; what technologies were used in the study; what devices were used; what vital signs were measured; and how common the method was compared to other studies.

### 3.1. Performance Metrics

For the purpose of evaluating the model and comparing it to the articles of other authors, several common performance indicators were employed. This section provides the specifics of the most commonly used metrics and their derived methods. Only the metrics that were listed in tables including evaluated papers and that were utilized by more than five (inclusive) articles will be detailed. The effectiveness of the proposed approach was assessed using a number of quality criteria, such as the mean error (ME), standard deviation (STD), mean absolute error (MAE), root mean square error (RMSE) and mean absolute percentage error (MAPE). The metrics shown in Table 2 represent the error between the estimated (es) XR and the ground truth (lb) values, where X ∈ [H (heart), R (respiratory)].

Performance metrics were analyzed in all articles, and it was found that 90% of articles could be compared by using the MAE, while less than 70% of articles, with RMSE. The unit of beats per minute or bmp is often used to represent the physical difference between estimated and ground truth heart rates (HRs). The breaths per minute or bmp is often used to represent the physical difference between estimated and ground truth respiratory rates (RRs). The mean absolute percentage error is represented with units of percentage (%). The units of millimeters of mercury or mmHg is used to measure differences in other vital signs such as the systolic blood pressure (SBP) and diastolic blood pressure (DBP). Blood oxygen saturation level is measured in percentage (%) units. The following Table 3 provides summaries of reviewed articles filtered by the subject “heart rate”.

As seen from Table 3, using the rPPG, it was possible to obtain data accuracy close to 5 bpm from certified or tested contact-based photoplethysmography using common datasets. By using custom or their own datasets, article authors claimed to obtain an accuracy of 1–2 bpm, which is truly remarkable, but based on the dataset size, it is obvious that the authors used a personalized dataset and personalized results. In common datasets with higher variability, results were less accurate for the whole array of tests.

For blood pressure estimation from rPPG, custom datasets were mostly used. Table 4 displays the filtered results. In summary, the outcomes of tests in this area were unknown or hidden. Of the published results, the highest accuracy was 75–80%. This is unusable in medical testing. By medical device usage certification, there were three levels:adequate for a “high-accuracy” device (defined as resulting in a mean BP difference between the reference and test device measurement and its associated standard deviation of 0 ± 3–6 [mean ± SD] mmHg), as it would have <14% chance to fail;inadequate for a “moderate accuracy” device (difference of 4 ± 5 mmHg), as it would have 28% of a chance to fail, which is unacceptably high;adequate for a “low accuracy” device (difference of 6–8 ± 5 mmHg, or 0 ± 10–12 mmHg, or 4–6 ± 8 mmHg), as it would have 94% chance to fail.

**Table 4 sensors-24-03963-t004:** Summary of papers in non-contact vital sign monitoring analyzing blood pressure.

Reference	Methodology	Data Set	Results
Das B. et al., 2022 [1]	remote photoplethysmography (rPPG)	MIMIC III 53,423 of subjects	-
Steinman et al., 2021 [2]	remote photoplethysmography using smartphone and cameras	-	-
Gao et al., 2023 [43]	remote photoplethysmography (rPPG) long short-term memory network (LSTM)	IIP-HCI dataset UBFC-Phys dataset LGI-PPGI dataset Custom dataset	-
Van Putten et al., 2023 [44]	remote photoplethysmography (rPPG) 100 discriminating feauters	Custom dataset (4500 measurement)	Acc = 79% of all individuals with hypertension
Wu et al., 2023 [45]	remote photoplethysmography (rPPG) Windkessel model and hand-crafted waveform characteristics	Chiao Tung BP (CTBP) dataset	MAE_SBP_ = 6.48 mmHg MAEDBP = 5.06 mmHg
Bousefsaf et al., 2022 [46]	imaging photoplethysmographic (iPPG) deep U-shaped neural network	BP4D+ (140 participants) Custom dataset 57 participants based-on AMM and BHS	MAE_DBP_ = 5.1 mmHg; MAESBP = 6.73 mmHg
Wiffen et al., 2023 [28]	developing a measurement protocol	Custom dataset with patients data 1950 participants	-
Qiao et al., 2022 [32]	web-camera based solution	PURE dataset	MAE_HR_ = 1.73 bpm MAEHRV = 18.55 ms MAESpO_2_ = 1.64%
Wuerich et al., 2022 [47]	remote photoplethysmography (rPPG)	Custom dataset	MAE_SBP_ = 5.5 ± 4.52 mmHg MAEDBP = 3.7 ± 2.86 mmHg
Schrumpf et al., 2021 [48]	remote photoplethysmography (rPPG) deep learning techniques	MIMIC-III (12000 records of PPG)	MAE_AlexNet_ = 15.7 SBP mmHg MAEREsNet = 13.02 SBP mmHg
Shirbani et al., 2021 [49]	video-based photoplethysmography (vPPG)	Custom dataset (10 subjects video records of face and palm)	PTT-BP Correlation coefficient = 0.8
Tran et al., 2020 [42]		Custom datset	RMSE_SBP_ = 7.942 RMSEDBP = 7.912
Fan et al., 2020 [50]	remote photoplethysmography (rPPG) Gaussian model	Custom dataset	r_SBP_ = −0.84 rDBP = −0.66

From the results of reviewed articles, the closest results on customer datasets had a moderate accuracy, which could be used in preliminary BP testing but not final health estimation.

For remote blood pressure monitoring, there were the lowest-accuracy results; on the other hand, for SpO_2_ monitoring, there were quite high-accuracy results. In Table 5, the represented results of SpO_2_ estimation from various articles are shown. It can clearly be seen that oxygen saturation could be measured with a 1.64% error or higher than 95% accuracy level using standard datasets like PURE or large customer datasets, as in Wiffen et al.’s article [28].

The majority (around 80%) of research articles focused on remote heart rate estimation and a bit less on remote respiratory rate estimation. Less than 20% of research focused on remote blood pressure monitoring. Remote photoplethysmography (rPPG) technology served as the basis for remote monitoring of vital signs. Measuring technologies such as video-based photoplethysmography (vPPG), video plethysmography (VPG) and imaging photoplethysmography (iPPG) refer to the same remote monitoring technique. Respiratory rate, heart rate, heart rate variability and oxygen saturation were estimated by measuring the skin surface of the subject’s face. The pulsatile pressure wave was measured in the areas of the neck artery or the palmar artery. Blood pressure, both systolic and diastolic, was estimated by focusing image sensors into two body areas that were at a certain distance from each other, such as, the forehead and palm. Estimated values of vital signs were compared with values that were acquired using contact diagnostic devices, such as, electrocardiographs, finger blood pressure monitors, etc. Current solutions, which rely on remote PPG analysis, were adapted to specific scenarios by utilizing limited public and/or custom datasets. There is a lack of research on the deep learning-based vital signs used to estimate a model’s resilience and capacity for generalization.

### 3.2. Datasets

Although deep learning models reduce the need for manual feature engineering, they increase the number of model parameters. These kinds of models require large datasets for training, and while public, general-purpose datasets have accumulated sufficient samples, the availability of publicly available medical data restricts the applications of deep learning. Inference time constraints and the hardware a model will run on also limit a model’s development. Ethical considerations restrict the acquisition of medical pictures, and the absence of retrospective patient approval constricts the use of already-existing imagery. In this setting, scientists are encouraged to work together with healthcare facilities and other researchers to obtain new annotated samples or to repurpose reliable, openly accessible data. This section aims to account for major publicly available data by listing datasets reported in the reviewed research.

Table 6 lists all datasets that were made publicly available. We identified some limitations and shortcomings of the aforementioned rPPG datasets, and some of them appeared as follows. The majority of publicly available datasets provided labels for heart rate and respiratory rate only. The MTHS dataset contained an acceptably sufficient number of samples (enough to train a neural network-based classifier) but provided labels for HR and SpO_2_ only. More significantly, the availability of only a few publicly accessible video-PPG datasets would limit the ability to evaluate the generalization capacity, accuracy and resilience of any deep learning system for vitals estimation. This is why a new and sizable dataset on video-PPG is required. The majority of available datasets used a straightforward video-collecting technique. For example, samples within a single set were gathered under constant conditions, and incorporating videos from other datasets might have helped improve dataset variety. Diverse patient ethnicities, lighting settings and recording equipment contribute to a variety that is advantageous for a classification model intended for usage by non-professionals with commercially accessible smartphones at home.

### 3.3. Architectures of Regression Models

Vital sign estimation based on deep neural networks is a commonly used backbone term to describe the regression part of state-of-the-art articles. Vital sign monitoring plays a crucial role in healthcare, and AI-based regression models in this field are more accurate and efficient. These models estimate vital signs (heart rate, blood oxygen saturation, etc.) by analyzing various signals like remote photoplethysmography (rPPG). The most common and prominent approach utilizes deep learning architectures, particularly convolutional neural networks (CNNs). These models process an input signal through a series of convolutional layers, extracting features that correlate with vital signs. Techniques like residual connections and a discrete cosine transform (DCT) and other transformations are incorporated to enhance feature extraction. Deep neural networks (DNNs), particularly convolutional neural networks (CNNs), dominate the field. A major challenge is the limited size of datasets in vital sign estimation research. Training DNNs effectively often requires vast amounts of data. A survey of articles indicates that the size of the datasets, which are small, limits researchers to mostly relying on pretrained, known feature extraction and regression architectures. To address this issue, researchers leverage pretrained models, typically trained on large, generic datasets for image recognition or other tasks. These pretrained models act as powerful feature extractors, capturing essential patterns within the data. Subsequently, these features are finetuned for the specific task of vital sign estimation.

While pretrained models offer a solid foundation, some researchers opt for custom convolutional neural networks (CNNs) in combination with data transformations. This approach allows for greater control over feature extraction specific to the vital signs of interest. Data transformations, like discrete cosine transform (DCT), can further enhance a model’s ability to identify relevant patterns in its input signals. The custom CNN approach is particularly appealing when there is a need for increased efficiency. By carefully designing network architectures and potentially reducing the number of parameters, researchers are developing models suitable for deployment on mobile devices or resource-constrained environments.

Another area of focus is designing compact and efficient models suitable for deployment on mobile devices. This enables continuous monitoring and promotes accessibility. Researchers achieve this by reducing the number of parameters in a model or employing lightweight architectures. Researchers are actively exploring ways to improve both accuracy and efficiency using lightweight architectures, transfer learning and explainable AI., including the following factors: designing models with fewer parameters for deployment on resource-constrained devices; leveraging knowledge gained from estimating one vital sign to improve the estimation of others; and developing models that explain their reasoning behind vital sign estimations, fostering trust and improving clinical decision-making. AI-based vitals estimation models offer a promising technique. The choice of architecture depends on the desired balance between accuracy and computational efficiency. Researchers strive to develop AI-based regression models that are not only accurate but also efficient and interpretable for vital sign estimation, paving the way for wider adoption in healthcare settings.

## 4. Future Research

If an rPPG signal were accurate and reliable to acquire through not only BPM readings but also cardiogram trends like from an ECG, this would further prove the efficacy of using this method in clinical applications. With the current discussed method, there is the possibility of measuring heart rate variability (HRV). In recent research, the conclusion was made that HRV is heavily related to autonomic nervous system functionality. HRV depression has been observed in various clinical situations, such as autonomic neuropathy, heart transplantation, congestive heart failure, myocardial infarction (MI) and other cardiac and noncardiac diseases. It is crucial to understand that the clinical significance of HRV analysis has only been clearly acknowledged in two specific clinical situations: as a means of predicting the risk of arrhythmic events or sudden cardiac death following an acute myocardial infarction, and as a clinical indicator of developing diabetic neuropathy. More recently, its significance in the assessment and treatment of heart failure has also been acknowledged. It is crucial to acknowledge the constraints of HRV in terms of its current clinical usefulness. The standardization of HRV methodology has been inadequate [136].

Looking forward in terms of research possibilities, by obtaining reliable rPPG readings in the form of oscillometric pressure pulse waves, this would open up even more medical information about the tested person, like their history of arrythmia or blood pressure [137]. In the overviewed PPG articles, there were attempts to obtain blood pressure by measuring the time between the heart cycle start, pulse peak and heart cycle end, like in Figure 6. 

In such a way, using mathematical calculations, it is possible to obtain a somewhat estimated blood pressure. By integrating deep learning techniques, it is possible to improve the accuracy of readings. However, as of the publication of this article, the results obtained are still below the acceptable medical threshold when considering the universally accepted criteria for validating blood pressure measuring devices [138]. 

## 5. Conclusions and Discussion

For most remote vital sign monitoring articles, video and RGB image processing using cameras were mentioned, where each color range extracts a specific part of an image difference based on blood vessels’ proximity to outer skin. From exploring the color of outer skin, it was noticed that the closer the blood vessel is to it, the more bluish the color of the outer skin becomes. By employing image recognition and mathematical calculation as well as AI algorithms, it is possible to compare video frames and from them, by processing differences in color, it is possible to determine pulse. As blood pressure changes, so does vessel diameter change and so does its closeness to outer skin. Respiratory rate was measured by changes in body movements. It is the easiest vital sign to measure, as even if a human is covering their face with a mask, it is possible to check their breathing rate by mask or bodily micro-movements. Based on the examined articles, the accuracy of measurement of vital signs using remote photoplethysmography methods depends on factors including human skin color, ambient light signs taken from the face and the direction of the human gaze. By combining camera image processing using algorithms with additional convolutional neural network (CNN) processing, the reliability of data is increased dramatically. Additional light can be introduced, if possible, to increase accuracy.

There was a lack of deeper research on blood pressure (BP) and the methods and possibilities around its use. Most of the articles that did mention it only did so in the results of their investigation, and not in the methods themselves. There is a need for open-source databases and code availability to be approachable by more scientific studies and universities; that way, there will be an increased amount of research and articles with more reliable methods. Further investigation is required regarding remote blood pressure measurements, specifically to address concerns regarding accuracy, validation in various environments, long-term reliability, repeatability, and user behavior comprehension. The accuracy of remote methods such as video-based photoplethysmography has to be compared with traditional arm cuffs, which are the gold standard for blood pressure measurement. While some research shows promising results, larger and more diverse populations need to be included to ensure accuracy across various demographics and health conditions. The majority of experimental investigations were executed in controlled clinical settings. However, real-world use at home or in remote areas might introduce variations due to user techniques, environmental factors or device limitations. More research is needed to see how these remote-sensing technologies perform in everyday scenarios. State-of-the-art research often focuses on short-term use. Extended monitoring over weeks or months is needed to assess if remote technologies provide consistent and reliable readings. Remote monitoring might capture average blood pressure, but it might miss important fluctuations that traditional cuffs can detect. More research is needed to understand if remote technologies can effectively capture this variability. The success of remote monitoring depends on user adoption and adherence. Research on user behavior can help identify factors that influence compliance and develop strategies to improve it.

These are just some of the areas where deeper research is needed for remote blood pressure measurements. Nowadays, every article covers its own approach to problem solving, and most of them are theoretically based on simulations like MATLAB’s Simulink or Python, and not on real-world applications. While simulations are valuable tools, the overemphasis on theoretical approaches in articles can lead to a gap between theory and practice. Simulations enable researchers to isolate and manipulate variables to understand their individual and combined effects on a system. This level of control can be difficult to achieve in real-world experiments. Simulations allow for a quick exploration of multiple design options. Researchers can test different configurations and identify the most promising ones efficiently. Many articles do not however provide enough detail on how to translate simulated solutions into real-world implementations and how their proposed method was validated in real-world settings. Real-world systems are complex and can be influenced by unforeseen factors not captured in a simulation. Simulations might miss crucial details that could impact the final outcome. Findings from simulations conducted on specific scenarios might not be generalizable to broader real-world applications. 

## Figures and Tables

**Figure 1 sensors-24-03963-f001:**
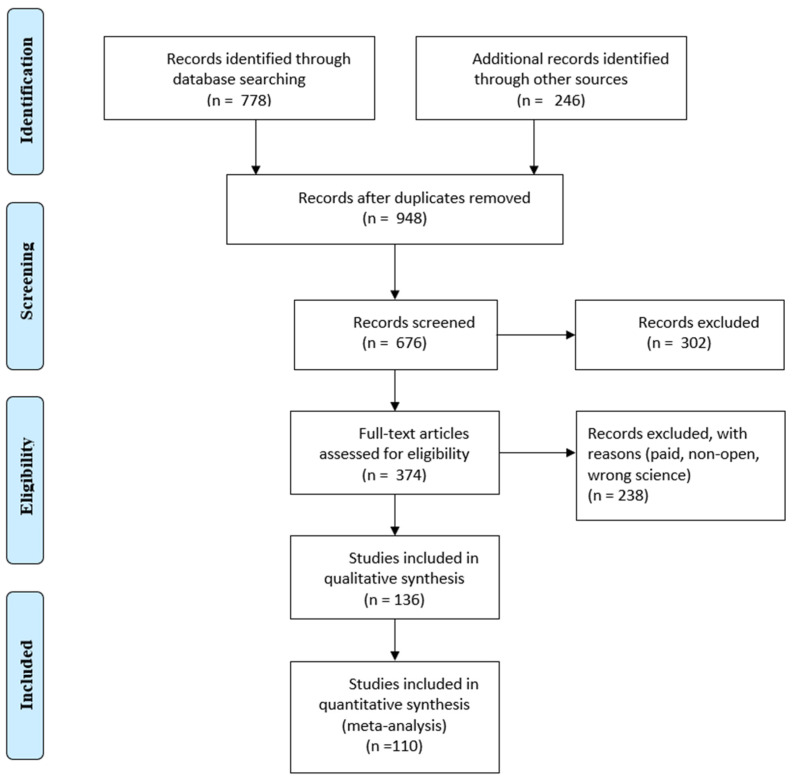
PRISMA flow diagram.

**Figure 2 sensors-24-03963-f002:**
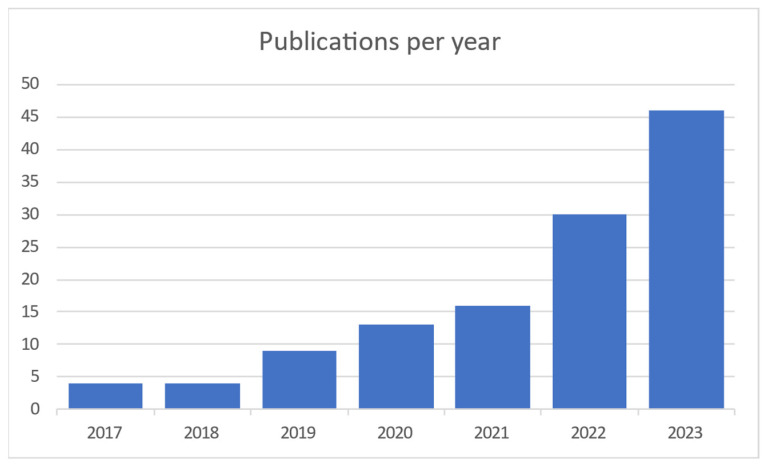
Publications per year [4,5,6,7,8,9,10,11,12,13,14,15,16,17,18,19,20,21,22,23,24,25,26,27,28,29,30,31,32,33,34,35,36,37,38,39,40,41,42,43,44,45,46,47,48,49,50,51,52,53,54,55,56,57,58,59,60,61,62,63,64,65,66,67,68,69,70,71,72,73,74,75,76,77,78,79,80,81,82,83,84,85,86,87,88,89,90,91,92,93,94,95,96,97,98,99,100,101,102,103,104,105,106,107,108,109,110,111,112,113,114,115,116,117,118,119,120,121,122].

**Figure 3 sensors-24-03963-f003:**
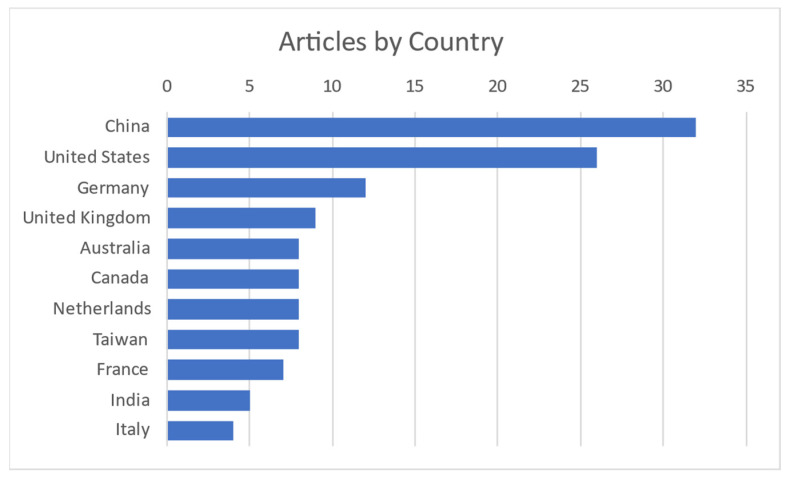
Publications per country [4,5,6,7,8,9,10,11,12,13,14,15,16,17,18,19,20,21,22,23,24,25,26,27,28,29,30,31,32,33,34,35,36,37,38,39,40,41,42,43,44,45,46,47,48,49,50,51,52,53,54,55,56,57,58,59,60,61,62,63,64,65,66,67,68,69,70,71,72,73,74,75,76,77,78,79,80,81,82,83,84,85,86,87,88,89,90,91,92,93,94,95,96,97,98,99,100,101,102,103,104,105,106,107,108,109,110,111,112,113,114,115,116,117,118,119,120,121,122].

**Figure 4 sensors-24-03963-f004:**
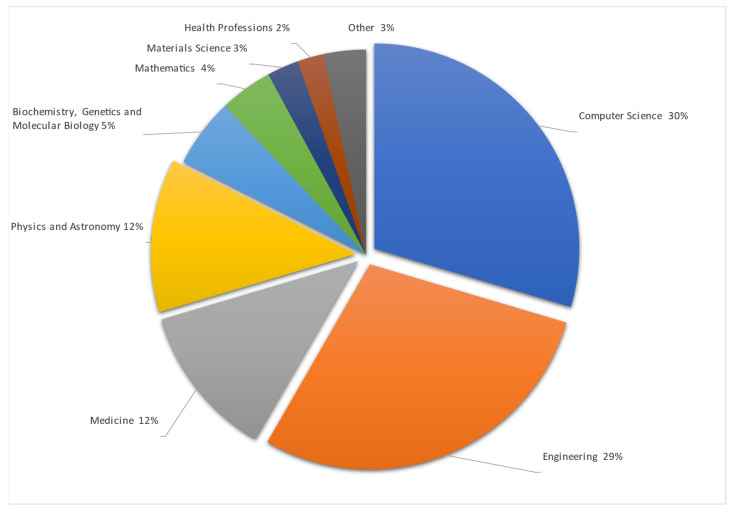
Publications by subject area [4,5,6,7,8,9,10,11,12,13,14,15,16,17,18,19,20,21,22,23,24,25,26,27,28,29,30,31,32,33,34,35,36,37,38,39,40,41,42,43,44,45,46,47,48,49,50,51,52,53,54,55,56,57,58,59,60,61,62,63,64,65,66,67,68,69,70,71,72,73,74,75,76,77,78,79,80,81,82,83,84,85,86,87,88,89,90,91,92,93,94,95,96,97,98,99,100,101,102,103,104,105,106,107,108,109,110,111,112,113,114,115,116,117,118,119,120,121,122].

**Figure 5 sensors-24-03963-f005:**
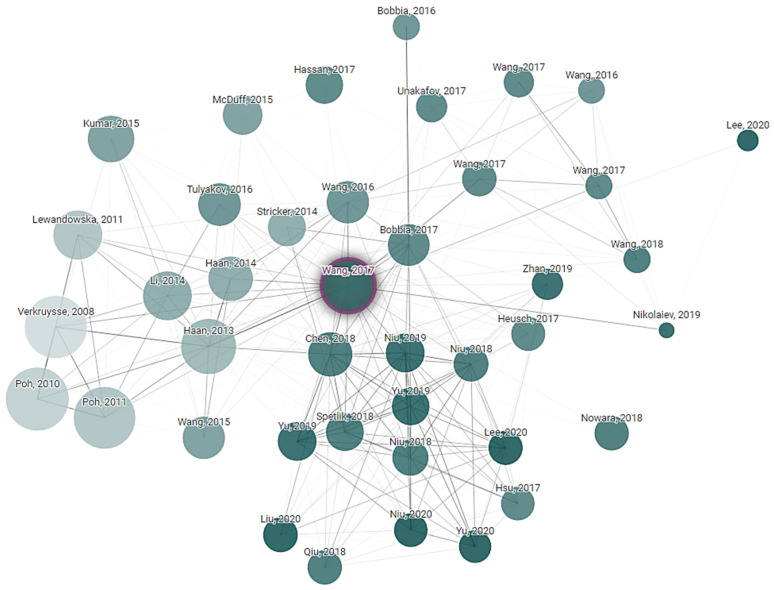
Article dependency network of research that explored the development and advancements in non-contact vital sign monitoring technologies.

**Figure 6 sensors-24-03963-f006:**
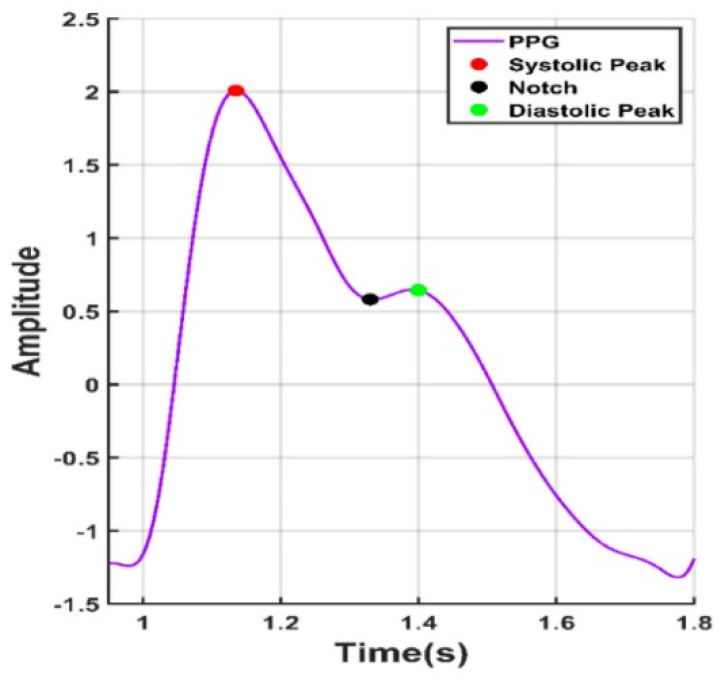
Points of interest from a PPG signal.

**Table 1 sensors-24-03963-t001:** Article eligibility criteria.

Inclusion Criteria	Exclusion Criteria
Published in English	Business
Journal articles	Veterinary Medicine
Full text available	Arts and Humanities
Article is published	Agriculture and Biological Science
Year of study between 2016 and 2024	Chemical engineering
	Chemistry
	Immunology and Microbiology
	Economics
	Pharmacology
	Earth and Planetary Science
	Psychology
	Decision Science

**Table 2 sensors-24-03963-t002:** Model performance metrics.

Metric	Formula	Description
Mean error (ME)	$ME=\frac{1}{N}\sum_{t=1}^{N}\left({XR}_{lb}\left(t\right)-{XR}_{es}\left(t\right)\right)$	Mean error between the estimated and ground truth signal
Standard deviation (STD)	$STD=\sqrt{\frac{1}{N}\sum_{t=1}^{N}{\left({XR}_{es}\left(t\right)-ME\right)}^{2}}$	Standard deviation between the estimated signal and ground truth
Mean absolute error (MAE)	$MAE=\frac{1}{N}\sum_{t=1}^{N}\left|{XR}_{lb}\left(t\right)-{XR}_{es}\left(t\right)\right|$	Mean absolute error between the estimated and ground truth signal
Root mean square error (RMSE)	$RMSE=\sqrt{\frac{1}{N}\sum_{t=1}^{N}{\left({XR}_{lb}\left(t\right)-{XR}_{es}\left(t\right)\right)}^{2}}$	Root mean square error between the estimated and ground truth signal
Mean absolute percentage error (MAPE)	$MAPE=\frac{1}{N}\sum_{t=1}^{N}\left|\frac{{XR}_{lb}\left(t\right)-{XR}_{es}\left(t\right)}{{XR}_{lb}\left(t\right)}\right|$	Mean absolute percentage error between the estimated and ground truth signal

**Table 3 sensors-24-03963-t003:** Summary of papers in non-contact vital sign monitoring analyzing heart rate.

Reference	Methodology	Dataset	Results
Song et al., 2020 [4]	remote photoplethysmography super-high resolution imaging up to 2704 × 1520 pixels	public UBFC-RPPG (43 videos of 42 different subjects) self-collected video (15 videos of 15 different individuals)	1 m distance, highest resolution, r = 0.993; 3 m distance, highest resolution, r = 0.863;
Cheng et al., 2021 [5]	remote photoplethysmography deep learning techniques	-	-
Mehmood et al., 2023 [6]	CNN based model	Welltory dataset (21 video and ECG recordings)	MAE_HR_ = 5.91 MAESpO_2_ = 2.01 MAERR = 3.11
Tong et al., 2023 [7]	Imaging Photoplethysmography (IPPG) remote RGB observations Biorthogonal wavelet decomposition	UBFC-rPPG dataset Real scenario (7 individuals)	Correlation coefficient = 0.92
Duan et al., 2023 [8]	Imaging Photoplethysmography (IPPG)	PURE VIPL-HR UBFC-RPPG MAHNOB-HCI	MAPE = 4.65 RMSE = 4.17 MAE = 3.45
Lin et al., 2023 [9]	remote photoplethysmography convolutional neural networks	-	-
Karthick et al., 2023 [10]	remote photoplethysmography (rPPG)	Custom dataset	-
Qayyum et al., 2023 [11]	remote photoplethysmography (rPPG)	Custom dataset 18 participants	-
Molinaro et al., 2023 [12]	single digital camera	Custom dataset	MAE < 5 bpm MAE < 3.42 bpm
Ouzar et al., 2023 [13]	imaging photoplethysmography end-to-end spatio-temporal network (X-iPPGNet)	BP4D+ MAHNOB-HCI UBFC-rPPG MMSE-HR	MAE_HR_ = 4.10 MAEUBFC = 4.99 MAEMAHNOB = 3.17
Zhang et al., 2023 [14]	remote photoplethysmography (rPPG)	Custom dataset	Acc = 94.5%
Liu et al., 2023 [15]	remote photoplethysmography (rPPG) convolutional neural network	MAHNOB-HCI PURE	MAE = 3.12 bmp SD = 3.78
Firmansyah et al., 2023 [16]	remote photoplethysmography (rPPG) 1D convolutional neural network	Custom dataset (10 subjects) Custom dataset (10 subjects)	MAE = 2.78 bpm
Smiley et al., 2023 [17]	Image-based photoplethysmography (iPPG)	-	-
Guler et al., 2023 [18]	remote photoplethysmography (rPPG) reducing signal-to-noise ratio noise filter selection	Custom dataset (300 recordings)	-
Ontiveros et al., 2023 [19]	remote photoplethysmography (rPPG) contact-based photoplethysmography (cPPG)	Custom dataset	-
Shenoy et al., 2023 [20]	Imaging photoplethysmography (iPPG) unrolling proximal gradient descent	MMSE-HR	MAE = 1.11 bmp RMSE = 2.97 bmp
Zhalbekov et al. 2023 [21]	remote photoplethysmography (rPPG) blood volume pulse (BVP)	LGI dataset	MAE = 5.39 bmp
Xu et al., 2023 [22]	illumination variation robust remote-photoplethysmography (Ivrr-PPG)	Custom (NIR light source 940nm)	RMSE_1_ = 2.94 RMSE4 = 9.11
Hu et al., 2023 [23]	Photoplethysmography (PPG)	Custom dataset	-
Revanur et al., 2023 [24]	Video-based physiological signal estimation	Vision-for-Vitals (V4V) benchmark	MAE = 13.0 bmp
Hu et. al., 2022 [25]	remote photoplethysmography (rPPG) spatial-temporal attention network	PURE MMSE-HR UBFC-rPPG	MAE = 0.23 RMSE = 0.48 R = 0.99
Jorge et al., 2022 [26]	non-contact camera-based monitoring	Custsom	MAE_HR_ = 2.5 bmp MAERR = 2.4 bmp
Wang et al., 2022 [27]	remote photoplethysmography (rPPG) anti-motion interference method T-SNE-based signal separation (TSS)	UBFC-RPPG VIPL-HR	MAE_UBFC_ = 1.64 bpm MAEVIPL = 4.76 bpm
Wiffen et al., 2022 [28]	remote photoplethysmography (rPPG)	Custom dataset with patients data 1950 participants	-
Przybyło, 2021 [29]	Video-plethysmography (VPG) Long Short Term Memory (LSTM)	Custom	MAE = 3.62 bpm
Liu, 2022 [30]	remote photoplethysmography (rPPG) Face tracking and SVM	PURE	MAE = 2.52 bpm
Han et al., 2022 [31]	structural sparse representation method to reconstruct the pulse signals structural sparse representation method to reconstruct the pulse signals	UBFC COHFACE	MAE = 2.57 bpm
Qiao et al., 2022 [32]	remote photoplethysmography (rPPG) web-camera based solution	TokyoTech rPPG PURE	MAE_HR_ = 1.73 bpm MAEHRV = 18.55 ms MAESpO_2_ = 1.64%
Sun et al., 2022 [33]	remote photoplethysmography (rPPG) Unsupervised learning	PURE UBFC-rPPG	MAE_UBFC_ = 0.64 bpm MAEPURE = 1 bpm MAEOBF = 0.51 bpm
Ding et al., 2022 [34]	remote photoplethysmography (rPPG) multi-physiological signals estimation network (SMP-Net) based on multimodal fusion	multi-vital sign (MMVS) VIPL-HR	MAE_HR_ = 1.12 bpm MAERR = 2.08 bpm
Das et al., 2022 [35]	remote photoplethysmography (rPPG) spatial-temporal filtering method	Custom dataset (25 persons) COHFACE (160 videos 40 persons)	RMSE_COHFACE_ = 2.41 RMSEcustom = 0.82
Zheng et al., 2022 [36]	remote photoplethysmography (rPPG) Compensation algorithm of ambient light and body motions	Custom dataset with facial videos	MAE = 4.32 bpm
Abbas et al., 2021 [37]	remote photoplethysmography (rPPG)	Custom dataset (75 volunteers)	MAE_HR_ = 10 bpm MAERR = 4 bmp
Ryu et al., 2021 [38]	remote photoplethysmography (rPPG) singular spectrum analysis and sub-band	UBFC-RPPG dataset	correlation coefficient = 0.89
Kado et al., 2020 [39]	remote photoplethysmography (rPPG) Spatial-Spectral-Temporal Fusion	Custom dataset	MAE < 5 bpm
Zhang et al., 2020 [40]	remote photoplethysmography (rPPG) 3D model-based compensation algorithm of motion artefacts and varying lightening	Custom dataset that employs multimodal 3D imaging system	-
Liu et al., 2020 [41]	video-based monitoring drastic facial unsteadiness disturbance-adaptive orthogonal matching pursuit (DAOMP) algorithm	Custom dataset (268 subjects for training) (67 subjects for testing) MAHNOB-HCI	MPE_custom_ = 1.55 bpm MPEMAHNOB = 1.26 bpm
Tran et al., 2020 [42]	remote photoplethysmography (rPPG)	Custom datset	RMSE_SBP_ = 7.942 RMSEDBP = 7.912

**Table 5 sensors-24-03963-t005:** Summary of papers on non-contact vital sign monitoring analyzing SpO_2_.

Reference	Methodology	Dataset	Results
Mehmood et al., 2023 [6]	Model based on visual transformers	MTHS dataset (62 videos and SpO_2_ recordings)	MAEHR = 5.91 MAESpO_2_ = 2.01 MAERR = 3.11
Wu et al., 2023 [51]	Color camera	Custom dataset (60 subjects) Recorded with mobile phone, webcam and industrial camera	MAEphone = 4.39 MAEwebC = 4.45 MAEcamera = 4.22
Wu et al., 2023 [51]	K-nearest Neighbor (KNN)	Custom dataset (60 subjects) Recorded with mobile phone, webcam and industrial camera	MAEphone = 4.39 MAEwebC = 4.45 MAEcamera = 4.22
Qayyum et al., 2023 [11]	Remote photoplethysmography (rPPG)	Custom dataset (18 participants)	–
van Gastel et al. (2022) [52]	Camera-based measurements	Custom dataset	Accuracy of normal SpO_2_ level > 95% Accuracy of low SpO_2_ level < 90%
Wiffen et al., 2022 [28]	Developing a measurement protocol	Custom dataset with patient data of 1950 participants	–
Qiao et al., 2022 [32]	Webcam-based solution	PURE dataset	MAEHR = 1.73 bpm MAEHRV = 18.55 ms MAESpO_2_ = 1.64%

**Table 6 sensors-24-03963-t006:** Available datasets for rPPG research.

Database	No. of Subjects	No. of Videos	Data Type	Task/Conditions	Ethnicity
BP4D+ [128]	140	1400	25 fps, 1040 × 1392 pixels, 3D, 2D, thermal and physiological data sequences	Emotion elicitation	Latino/Hispanic, White, African American, Asian and Others
MAHNOB-HCI [129]	27	527	61 fps, eye gaze, physiological sensors measuring ECG, EEG (32 channels), respiration amplitude and skin temperature	Emotion elicitation	Caucasian and Asian
UBFC-rPPG [130]	42	42	30 fps, 640 × 480 pixels, CMS50E transmissive pulse oximeter	Interaction	–
MMSE-HR [131]	40	102	25 fps, 1040 × 1392 pixels, blood pressure signal	Emotion elicitation	Latino/Hispanic, White, African American, Asian and Others
BUT-PPG [132]	12	48	30 fps, smartphone camera, ECG signals	Interaction	–
AFRL [133]	25	300	30 fps, 658 × 492 pixels, contact reflective PPG sensor	Interaction Head motion	–
NBHR [88]	257	1130	30 fps, photoplethysmograph information, heart rate and oxygen saturation level	Sleeping	infants at 0–6 days old
COHFACE [134]	40	160	20 fps, 640 × 480 pixels, heart rate and breathing rate	Interaction	–
VIPL-HR [135]	107	2378 (VIS) 752 (NIR)	Visible light and near-infrared light videos, BVP sensor data	Interaction	Caucasian and Asian
TokyoTech rPPG [81]	9	9	300 fps, 640 × 480 pixels, contact PPG sensor	Interaction	Caucasian and Asian

## Data Availability

Dataset available on request from the authors.

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
