# Peer review of "Non-Contact Vision-Based Techniques of Vital Sign Monitoring: Systematic Review"

_sensors, 2024, doi:10.3390/s24123963_

Round 1

Reviewer 1 Report

Comments and Suggestions for Authors

The current work is intended to be a Systematic review for Non-contact vision-based techniques of vital signs monitoring. As well as the structure of the work and presentation of the selection of works discussed on this article, the requirements to write a review are met. But as a concept for discussing these works it does not fulfill the requirement presented in the title of the work, namely, Systematic review for techniques.... That is, in this work on table 3 are presented a series of works from the literature of the last years that have as their subject non-contact vital signs monitoring. But the purpose of reviewing Non-contact vision techniques is not achieved, but rather it is a presentation of the field and of several works in this domain, in order to summarize the information of this field. A review involves the framing of works from various fields established by the author, the presentation of common points or the differences between them, the contribution of the author to the description of the respective field. In the present work, there is only a review of selected works from the chosen domain, without synthesizing the common information of the works.

Author Response

Thank you for reviewing our article. 

Best regards,

Prof. Raudonis

Reviewer 2 Report

Comments and Suggestions for Authors

This is an interesting report exploring the variety of approaches and outcomes related to video based physiological assessment. Yje guideline for review was followed precisely and articles appropriately chosen.  As just noted, the English expression is largely appropriate but some notably poor sentences and spelling errors occur, e.g. hearth for heart. Care must also be taken to ensure that definitions are present for abbreviations. An issue throughout and particularly for the ms title is whether 'vital signs' are the focus or blood pressure. Much of the text focusses on blood pressure but the majority of the work does not. The interesting core of the review is the variety of approaches and outcomes. The title might focus on this as a true evaluation of the validity of the techniques does not yet seem possible.The size of the main table is rather overwhelming. Some breakdown of this by a classification by,.e.g.,, methods used or outcome variable, might be helpful. Validity issues are mentioned briefly but might be further developed. Most particularly, a valid reading obtained from a comfortably seated patient in a light and temperature controlled room likely does not speak at all to validity in an ambulatory setting. Specific artifacts for specific variable might be noted,, e.g., variety in types of breathing, cardiac arrythmia, vascular reactions. Use of large databases will likely be useful for method development, but careful clinical/research studies must be done to establish validity.  

Comments on the Quality of English Language

This ms represents a considerable amount of effort and covers a great deal. The authors need to define their focus, however. They almost define their focus as blood pressure and then it turns out little can be said about this. They seem to be able to speak best to the variety of approaches and outcomes at present and how these illustrate further needs. 

A general ethical concern with the area might be potential covert use assessing one's health without their knowledge. 

Author Response

Thank you reviewing our article.

Best regards,

Prof. Raudonis

Round 2

Reviewer 1 Report

Comments and Suggestions for Authors

OK

Author Response

Thank you for reviewing.

Best regards,

Prof. Vidas Raudonis

Reviewer 2 Report

Comments and Suggestions for Authors

This revision is a clear improvement over the initial submission. The response to my critique were appropriate through not that expansive. Measurement issues are now noted but not expanded upon. This may be appropriate for the journal audience, but subsequent clinical and research studies are likely to be more critical of the limitation of video techniques. 

You may be interested in a forthcoming revision of the publication standards for heart rate and heart rate variability which is in press in the journal Psychophysiology.  The relative reliability/validity of EKG and photo measures are discussed.

Comments on the Quality of English Language

The language expression is reasonably clear, but some word usage and sentence structure could be improved. 

Author Response

(The authors gave the same response as above.)
